# The Effect of the Varietal Type, Ripening Stage, and Growing Conditions on the Content and Profile of Sugars and Capsaicinoids in *Capsicum* Peppers

**DOI:** 10.3390/plants12020231

**Published:** 2023-01-04

**Authors:** Carla Guijarro-Real, Ana M. Adalid-Martínez, Cherrine K. Pires, Ana M. Ribes-Moya, Ana Fita, Adrián Rodríguez-Burruezo

**Affiliations:** 1Biotecnología y Biología Vegetal, Escuela Técnica Superior de Ingeniería Agronómica, Alimentaria y de Biosistemas (ETSIAAB), Universidad Politécnica de Madrid, 28040 Madrid, Spain; 2Instituto de Conservación y Mejora de la Agrodiversidad Valenciana (COMAV), Universitat Politècnica de València, Camino de Vera s/n, 46022 Valencia, Spain; 3Centro Multidisciplinar, Universidade Federal do Rio de Janeiro, Campus UFRJ-Macaé, Macaé 27930-560, Brazil

**Keywords:** breeding, HPLC, organic farming, fructose, glucose, capsaicin

## Abstract

Peppers (*Capsicum* sp.) are used both as vegetables and/or spice and their fruits are used in a plethora of recipes, contributing to their flavor and aroma. Among flavor-related traits, pungency (capsaicinoids) and lately volatiles have been considered the most important factors. However, the knowledge of sugars is low, probably due to the fact peppers were historically considered tasteless. Here, using HPLC, we studied the content and profile of major sugars and capsaicinoids in a comprehensive collection of varietal types (genotype, G), grown under different growing systems (environment, E) in two years (Y) and considered the two main ripening stages (R). We found a major contribution to the ripening stage and the genotype in total and individual sugars and capsaicinoids. The year was also significant in most cases, as well as the G × E and G × Y interactions, while the growing system was low or nil. Ripening increased considerably in sugars (from 19.6 to 36.1 g kg^−1^ on average) and capsaicinoids (from 97 to 142 mg kg^−1^ on average), with remarkable differences among varieties. Moreover, sugars in fully ripe fruits ranged between 7.5 and 38.5 g kg^−1^ in glucose and between 5.2 and 34.3 g kg^−1^ in fructose, and several accessions reached total sugars between 40 and 70 g kg^−1^, similar to tomatoes. The results reveal the importance of the genotype and the ripening for these traits, particularly sugars, which should be considered key for the improvement of taste and flavor in peppers.

## 1. Introduction

*Capsicum* peppers are one of the most popular vegetables worldwide, with a yearly production of over 42 million t, cultivated across more than 3.7 million ha [1]. Originally from America, the cultivated peppers belong to five species, *C. annuum* being the most economically important one. Moreover, this species is phylogenetically close to other two cultivated species, *C. chinense* and *C. frutescens*, and the three species, together with their wild relatives, make up the *annuum–chinense–frutescens* complex or *annuum* complex [2]. *C. baccatum* and *C. pubescens* are the other two cultivated species, which belong to taxons that are different from the *annuum* complex, and their cultivation is mostly limited to the Andean region [2].

*Capsicum* peppers are appreciated as fresh vegetables, as well as a spice, providing color and characteristic aromas and flavors to a plethora of recipes, with variable levels of pungency [3]. In fact, their popularity as a spice favored its fast diffusion from America to the rest of the world since the 15th century [4]. The pungency of *Capsicum* peppers, usual in nature, is due to the accumulation of capsaicinoids that are synthesized in the fruit placenta, mainly capsaicin and dihydrocapsaicin [5,6]. The domestication and selection by farmers enabled the increase in pungency in cultivated forms, highly appreciated by many cultures, but also enabled the identification and breeding of sweet mutant peppers, carrying a deletion in the capsaicinoid synthetase gene [7]. Thus, capsaicinoid content in pepper fruits varies greatly from nil or minimal in non-pungent varieties to over 300 µg g^−1^ of fresh fruit (1000–5000 Scoville Heat Units, SHU) in common mild pungent *C. annuum* varieties [8]. Even more, the fully ripe fruits of *Bhut Jolokia* (a natural hybrid of *C. chinense* with introgressions from *C. frutescens*) or *Carolina Reaper* (a hybrid between *Bhut Jolokia* and *Habanero*), the most pungent peppers in the world, can accumulate more than 70 mg g^−1^ of dry fruit, reaching approximately between 1.2 and 3 × 10^6^ SHU [9,10].

Together with the importance of capsaicinoids, the studies on the volatile profile of peppers have increased considerably in the last decade, due to their organoleptic value [5,11,12,13,14,15,16]. By contrast, the importance of sugars in the taste and flavor of *Capsicum* fruits has been mostly neglected. Thus, the knowledge about the varietal diversity of sugars and their profiles and/or the impact of other agronomic effects are very scarce or nil, apart from a few studies (e.g., [6,17,18]). Sugars are the primary compounds related to the sweet taste. These compounds accumulate mainly in the pericarp of fruits and their levels may differ among cultivars, as found by Zamljen et al. [19], who determined up to 747 g kg^−1^ of dry weight (dw) in *C. chinense* × *C. frutescens* fruits but only 450 g kg^−1^ of dw on average in *C. annuum* varieties. Additionally, these metabolites, together with specific volatile compounds, can influence the whole flavor perception [20]. An interaction has been studied in other *Solanaceae* vegetables, such as tomatoes [21].

Although the accumulation of capsaicinoids and sugars in *Capsicum* peppers could be determined by the genotype, the environment where the plants grow or the ripening process can influence these levels as well. Environmental conditions vary among locations, cropping systems, and even years, including differences in factors such as soil composition and nutrient availability, the management of soil and irrigation, temperature and light intensity, and, in general, other abiotic stresses. Thus, some studies reported a combined effect of the genotype and location in the nutritional composition of *Capsicum* peppers [16,22,23,24], which can be remarkable in genotypes selected for specific local conditions, such as landraces. Lo Scalzo et al. [2] found an effect of the year of production for most agronomic and chemical traits in two sweet pepper varieties, and some of them, including the accumulation of sugars, were also affected by the growing system. Furthermore, the developmental stage of the fruit also affects the accumulation of metabolites, which usually increase to an optimal stage at the turning or the fully ripe stages [25,26], particularly sugars and antioxidants.

The present work was aimed at describing exhaustively the accumulation of sugars and capsaicinoids in a comprehensive selection of pungent and non-pungent *Capsicum* spp. varieties at the two main commercial stages of peppers, i.e., green-ripe and fully ripe fruits. Two years of cultivation were considered to determine the year effect on these accessions. Moreover, the evaluation across two different growing systems (conventional vs. organic management) was aimed at identifying whether the varieties were stable considering both systems or, on the contrary, they changed considerably between systems and, therefore, it would be necessary to select accessions adapted specifically to each system, i.e., exploiting the genotype × growing system interaction.

## 2. Results

### 2.1. Analysis of Variance for the Content of Sugars and Capsaicinoids

In general terms, the effects of genotype (G), ripening stage (R), and the year of cultivation (Y) were significant for the accumulation of both sugars and capsaicinoids, while the cultivation system (E) did not show a significant effect (Table 1). According to the percentage of the total sum of squares, the variation in total and individual sugars was mainly due to the ripening stage, followed to a lesser extent by the genotype and the year, whereas the genotype was the main factor contributing to the variation of capsaicinoids, followed to a very lesser extent by the ripening stage and the year effects (Table 1). Finally, the growing system did not appear to contribute significantly to most sugars and capsaicinoids, with the only exception being sucrose. In addition, overall significant interactions were found in most cases between the genotype and the rest of the factors, as well as between the ripening stage and the year (R × Y), which could bias the real magnitude of the contribution of some individual main factors. Finally, among the interactions involving the growing system, only G × E in all sugars and capsaicinoids, E × R in sucrose, E × Y in dihydrocapsaicin, nordihydrocapsaicin, and total capsaicinoids were found to contribute significantly to the variation of these traits, but at low level (Table 1).

In addition, to avoid biases involving the ripening stage, further ANOVA analyses considering separate green-ripe and fully ripe stages were also performed. In this regard, the genotype contributed greatly to the individual and total contents of both sugars and capsaicinoids at the green-ripe stage, i.e., 41–89% of the total sum of squares (TSS, Table 2). In addition, the G × Y interaction contributed significantly to the variation of sugars, with the only exception being sucrose (6.4–9.5% TSS), and individual and total capsaicinoids (5.6–21.7% TSS). Additionally, the G × E interaction was significant for the content of sugar, ranging between 2.5% (fructose) and 9.8% (total sugar), while the effect of this interaction in the variation of capsaicinoids was not significant or poorly significant. Finally, the E × Y interaction was only significant for the content of dihydrocapsaicin (0.24% TSS) (Table 2).

Regarding the fully ripe stage fruits, the genotype, the year effects, and their interaction were again the factors that mostly explained the variance of the studied traits (Table 2). In the case of sugars, the accumulated effects of these two factors and their interaction contributed to around 70% of the TSS for the variation of the total and individual sugars. Nevertheless, in comparison to the green-ripe stage, the contribution of the genotype to the TSS was considerably lower at the fully ripe stage (25–30% TSS), while the contribution of the year and the G × E and G × Y were higher (11.7–16.6%, 7.7–8.2%, and 29.2–30.4%, respectively) (Table 2).

On the contrary, in the case of capsaicinoids, the predominance of the genotype on the TSS was even higher at the fully ripe stage than at the green-ripe stage, i.e., between 61% (nordihydrocapsaicin) and 88.0% (capsaicin), followed by the effect of the G × Y interaction (4.7–21.8% for capsaicin and nordihydrocapsaicin, respectively) (Table 2). Finally, the growing system did not contribute significantly to the levels of either sugars or capsaicinoids, similar to the performance observed at the green-ripe stage, although there was a significant effect of the G × E interaction, which was higher in the case of sugars.

### 2.2. Sugar Content at the Green-Ripe and Fully Ripe Stages

#### 2.2.1. Total Sugar

At the green-ripe stage, the mean content for total sugars was 19.6 g kg^−1^ of fresh weight (fw). As indicated in the ANOVA, these values were mainly affected by the genotype and its interactions with the year and growing system (Table 2). This was confirmed by the broad range of variation for the content of total sugars between accessions (Figure 1). In year 1, Serrano accumulated more sugars than any other accession, i.e., a total of 30.2 g kg^−1^ and 37.6 g kg^−1^ fw under conventional and organic cultivation, respectively. BGV10582, Piquillo, Espelette, Guindilla, and Jalapeño showed values around 20–25 g kg^−1^ of fw, whereas Bola, Gernika, BOL58, and ECU994 showed medium to low content in total sugars, i.e., 10–15 g kg^−1^ of fw (Figure 1). In year 2, Serrano also accumulated high levels of total sugars in both growing systems (about 30 g kg^−1^ of fw). Jalapeno also reached high levels (25–30 g kg^−1^ of fw), particularly under organic cultivation. Accessions BGV10582, BOL58, and ECU994 again showed low levels, similar to those found in year 1, while sugars decreased considerably in Piquillo in the second year (~10 g kg^−1^ of fw reduction) (Figure 1). All these findings confirmed the significant G × Y interaction found in the ANOVA.

Finally, a G × E interaction was found in both years. Thus, the growing system affected significantly the levels of several accessions. As a result, the levels of total sugars differed between growing systems in many accessions or, alternatively, other accessions were found stable for this trait. Thus, in year 1, organic farming increased significantly the total sugars in Bola and Serrano peppers, while conventional farming increased the levels in Espelette, Guindilla, and ECU994, and the rest of the peppers in the growing system did not significantly change this trait. In year 2, total sugars decreased considerably under organic conditions in Bola and Gernika (by 2.7-fold and 4.5-fold, respectively), while their levels were maintained over systems in year 1 or even increased under organic conditions in the case of Bola (Figure 1 and Appendix A).

At the fully ripe stage, the average content of total sugars increased considerably in comparison to the green-ripe stage, reaching 36.6 g kg^−1^ of fw (a 17 g kg^−1^ average increase, *p* < 0.05; up to 87% higher) (Figure 2). Additionally, in agreement with the ANOVA, the accumulation of total sugars differed among genotypes, although the contribution of this factor was lower than that observed in the green-ripe stage, increasing the impact of other effects. Thus, the total sugars mean differed considerably between years, as the mean levels of year 1 were much higher than year 2 (42.7 g kg^−1^ of fw vs. 30.5 g kg^−1^ of fw, respectively, and *p* < 0.05), confirming the importance of the year effect (16.6% TSS) (Table 1). Moreover, the specific performance of several accessions differed between growing systems and particularly between years, confirming the remarkable effect of the G × E and G × Y interactions on this trait at the fully ripe stage. Thus, regarding the latter, accessions accumulating the highest levels in year 1 were Gernika, Jalapeño, Serrano, and BOL58, whose values were ≥50 g kg^−1^ (Figure 2). In the case of year 2, only Jalapeño reached similar levels to those found in year 1. Interestingly, BOL58 had the lowest accumulation of sugars in both conventional (10.0 g kg^−1^ of fw) and organic (8.7 g kg^−1^ of fw) systems, despite the high levels determined in year 1. Regarding the G × E interaction, the most illustrative examples were BGV10582, Bola, and Jalapeño, which showed higher accumulation of sugars under organic cultivation compared to the conventional system in both years, while sugars levels in Guindilla, Espelette, Gernika, and BOL58 showed differences between growing systems depending on the year (Figure 2).

#### 2.2.2. Sugar Profile

The ripening stage had also a remarkable effect on the profile of sugars. Thus, at the green-ripe stage, the three simple sugars, i.e., fructose, glucose, and sucrose were detected in the fruits, while by contrast, only fructose and glucose were detected in the fruits at the fully ripe stage (Figure 1 and Figure 2). Moreover, as observed in Figure 1, the contribution of each sugar to the profile, in the green-ripe peppers varied among the different accessions, while fully ripe peppers showed a more balanced contribution of both fructose and glucose (40–60% each) in most accessions (Figure 2).

At the green-ripe stage, as observed for total sugars, the profile of these metabolites was highly dependent on the genotype as said before, with a lower or nil effect of the growing system or the year (Figure 2). In this regard, some accessions, including Piquillo, Bola, Espelette, and Gernika, accumulated high percentages of sucrose (between 28% and 67% total sugar) in both years and growing seasons whereas, for the rest of the accessions, the contribution of sucrose to total sugars was lower than 20% (2–18%). Among these accessions, some of them showed a remarkable predominance of fructose against glucose, i.e., Jalapeño or Serrano (fructose > 50% of total sugar), while in others, both sugars showed more balanced percentages, comprised between 40 and 50% (Figure 2). Nevertheless, despite the low contribution of the growing system or the year, some differences due to these factors were found in specific accessions, which revealed the G × E and G × Y significant interactions detected in the ANOVA at this stage in most sugars (Table 2). Thus, for instance, we found remarkable differences (10–20% total sugar) between growing systems in the contribution of sucrose in Bola and Gernika in year 1, in Guindilla in year 2, and in Espelette in both years, or in the contribution of glucose in Bola in year 1 or Espelette and Guindilla in year 2 (Figure 1, Appendix A).

At the fully ripe stage, despite the significant quantitative differences among genotypes in total sugars (Figure 2) and in individual sugars (Appendix A), the contribution of both sugars to total sugars was very similar among genotypes (Figure 2), with a low or nil effect of the year or the growing system, as indicated in the ANOVA (Table 2). Thus, in year 2, fructose contributed between 51% and 58% to total sugars and the contribution of glucose was slightly lower, ranging between 42% and 49%, while no differences among growing systems were observed this year; even within each accession differences were very low or nil (Figure 2). In comparison, some little differences between genotypes and between growing systems within some accessions were found in year 1, in agreement with the significant contribution of the G × Y interaction and, at a lesser extent, G × E was detected in the ANOVA for these sugars at the fully ripe stage (Figure 2, Table 2). Thus, in year 1, the ranges of variation of sugars among accessions increased slightly in comparison to year 2, 43–61% in fructose and 39–57% in glucose, and accessions such as Bola, Espelette, Jalapeño, BOL58, or ECU994 reached fructose percentages >55% total sugars in both growing seasons, while in others such as Gernika or, particularly Serrano, the contribution of this sugar was ≤50% in both systems (Figure 2). Finally, in accessions such as BGV10582 and Guindilla, the organic system slightly increased (5–10%) the contribution of glucose to total sugars to the detriment of fructose, while in the rest of the accessions, no remarkable differences between systems were found (Figure 2, Appendix A).

### 2.3. Capsaicinoid Content at the Green-Ripe and Fully Ripe Stages

#### 2.3.1. Total Capsaicinoids

A broad range of variation for individual capsaicinoid content was found in the accessions tested, in agreement with the varietal diversity selected for this trial (pungent and non-pungent). We found from accessions with extremely low or nil capsaicinoids, i.e., the non-pungent sweet peppers BGV-10582, Piquillo, and Bola, to highly pungent accessions, such as the *C. chinense* ECU994, while the rest of accessions showed intermediate levels. Due to their lack of pungency, sweet accessions were discarded for further interpretation.

At the green-ripe stage, the mean content for total capsaicinoids was 95.7 mg kg^−1^ of fw, with a broad range of variation among accessions (Figure 3), confirming the predominance of the genotype effect on the variation of this trait, while the contribution of the growing system or the year was very low or nil. Nevertheless, some variations within the genotypes from year 1 to year 2 were also observed, confirming the considerable G × Y effect detected in the ANOVA (Table 2). Thus, ECU994 showed the highest levels among the accessions evaluated in both years and growing conditions (around 400–450 mg kg^−1^ of fw in year 1 and 300 mg kg^−1^ of fw in year 2, respectively). Serrano also accumulated remarkable levels, 130–150 mg kg^−1^ of fw, in both years and growing conditions. Finally, the rest of the accessions showed low–medium levels, Espelette and Guindilla being the most pungent accessions in this group (25–50 mg kg^−1^ of fw), with Gernika and Jalapeño showing levels < 10 mg kg^−1^ of fw (Figure 3). The *C. baccatum* accession showed an erratic performance between years with levels < 5 mg kg^−1^ in year 1 and close to 100 mg kg^−1^ of fw in year 2 (Figure 3). The most obvious effects of the G × Y interaction were the significant increases of total capsaicinoids in BOL58 from year 1 to year 2 and significant decreases in ECU994 from year 1 to year 2 in both growing seasons, while the rest of the accessions did not show great variations between years or growing seasons (Figure 3).

At the fully ripe stage, and similarly to sugars, the average content for total capsaicinoids increased to 143.3 mg kg^−1^ of fw (a 47 mg kg^−1^ increase, about a 50% increase in comparison to the green-ripe peppers). In fact, they increased in all the accessions, even in those with low levels. Additionally, as observed in the green-ripe peppers and advanced by the ANOVA, a broad range of variation was found among genotypes, with a low or nil average contribution of the other main factors, i.e., year and growing system (Figure 4). In addition, some significant and remarkable differences were found among years and, to a lesser extent, growing seasons within some accessions, revealing the presence of G × Y and G × E interactions (Table 2). Thus, total capsaicinoids ranged between the mild/low pungent Gernika and Jalapeño (<20 mg kg^−1^ of fw) and the highly pungent *C. chinense* ECU994 (400–700 mg kg^−1^ of fw), with the rest of accessions ranging between 50 and 200 mg kg^−1^ of fw (Figure 4). Regarding the G × Y interaction, we found some accessions which showed higher levels in year 1 than year 2, such as ECU994 (700 mg kg^−1^ of fw vs. 400–550 mg kg^−1^ of fw) and, to a lesser extent, Espelete, Guindilla, and Serrano, while other increased from year 1 to year 2, such as BOL58 or, even, did not experience significant changes between years, such as Gernika (Figure 4). Finally, in terms of G × E, some minor differences due to the growing conditions were found in Serrano in year 1 and year 2 (higher in conventional) and Guindilla and ECU994 in year 2 (higher in organic) (Figure 4).

#### 2.3.2. Capsaicinoids Profile

Regarding the profile of individual capsaicinoids, capsaicin, followed by dihydrocapsaicin, were the predominant compounds detected in most accessions, ripening stages, and growing conditions, accounting together for around 90% of the profile for nearly all accessions (Figure 3 and Figure 4). Nordihydrocapsaicin was the minor capsaicinoid (≤10% total capsaicinoids) and even in some cases it was not detected, e.g., Gernika at the green stage in both years and growing conditions, or BOL58 at the green-ripe stage in year 1 (Figure 3, Appendix A). Nevertheless, there were some differences in the profile of these metabolites due to the ripening stage, although lower than in sugar profiles. Additionally, some differences in the profile of capsaicinoids were found among accessions, as suggested in the ANOVA (Table 2), in particular the capsaicin:dihydrocapsaicin ratio.

Green-ripe fruits showed considerable differences among accessions. Thus, in accessions such as Espelette, Gernika, or particularly ECU994, the contribution of capsaicin was always ≥60%, reaching almost 100% in BOL58 in year 1 or Gernika in year 2, while others, such as Serrano, in both years or Jalapeño in year 1 showed a balanced accumulation of both capsaicin and dihydrocapsaicin (Figure 3). In addition, some differences among accessions were observed in the contribution of nordihydrocapsaicin, being almost nil on many accessions and years or reaching around 10% in accessions, such as Serrano, in both years and growing conditions, Jalapeño in year 1, or BOL58 in year 2 (Figure 3). By contrast, no differences were found between growing conditions. Finally, some variations between years were found within accession, while no differences were found between growing conditions, even within accessions, confirming the significant the G × Y interaction detected in the ANOVA and the low or nil contribution of the growing system and the G × E interaction. This was particularly obvious in Gernika, the 75:25 capsaicin:dihydrocapsaicin ratio and 100:0 in year 1 and year 2, respectively, or the 100:0 ratio of BOL58 in year 1, which became 55:35:10 (capsaicin:dihydrocapsaicin:nordihydrocapsaicin) in year 2, or Jalapeño, while other accessions such as Espelette, Serrano, or ECU994 were very stable between years in their capsaicinoid profile (Figure 3). Nevertheless, the erratic behavior observed in some genotypes could be due to their low levels of capsaicinoids (Figure 3, Appendix A).

In fully ripe fruits, the content of total capsaicinoids increased considerably in all the accessions, even in those with low levels. As found at the green-ripe stage, there were remarkable differences among the evaluated genotypes for the profile in capsaicinoids. At this stage, capsaicin was clearly the predominant capsaicinoid at levels ≥ 50% in most cases (45–83%), followed by dihydrocapsaicin (0–43%), and nordihydrocapsaicin (1–25%) (Figure 4). Additionally, the profiles observed at this stage in the different genotypes were quite similar to those found in most accessions at the green-ripe stage. Thus, accessions such as Gernika, BOL58, and, particularly, ECU994 in both years and BOL58 in year 1, showed again high proportions of capsaicin (60–83%) and low or almost nil levels of dihydrocapsaicin (>1–17%), while Serrano, Guindilla and, at a lesser extent, Espelette in both years and Jalapeño in year 1 showed more balanced proportions of capsaicin and dihydrocapsaicin (50% vs. 35–40%). Finally, differences in the contribution of nordihydrocapsicin were also found among accessions, BOL58 being the genotype with the highest proportions (12–28%), followed by Gernika (10–20%), Jalapeño, and Serrano, which showed levels of around 10–12% (Figure 4, Appendix A). In agreement with the ANOVA, global differences due to the growing conditions or the year were practically non-existent compared to the effect of the genotype, although some differences due to these effects were found within a few accessions, revealing a minor contribution of the G × Y and G × E interactions (Figure 4, Table 2 and Appendix A). In this regard, accessions such as Espelette, Gernika, Guindilla, Serrano, and ECU994 kept quite stable profiles between years, while Gernika, Jalapeño, and particularly BOL58 showed some differences due to this effect in the profile (Figure 4).

### 2.4. Effect of the Ripening Process on the Total Content of Sugars and Capsaicinoids

Our experiment revealed that the ripening process increased the levels of total sugars and total capsaicinoids, as in most accessions the levels in fully ripe fruits were similar or higher (≥slope 1) than those recorded in their equivalent green-ripe stage (Figure 5). Nevertheless, some differences were found between sugars and capsaicinoids. Thus, the increase due to the ripening was comprised between 1% and 100% (i.e., slope 1 and 2) in capsaicinoids in most cases, with very few accessions showing capsaicinoid levels lower at the fully ripe stage than the unripe stage (<slope 1) or higher than 100% (>slope 2) (Figure 5). By contrast, many accessions showed an increase in total sugars ≥ 100% (≥slope 2), particularly in accessions grown under organic conditions, while the increases found in conventional farming were mainly comprised between 1% and 100% (Figure 5).

Considering the years, our findings suggest that the increase in sugars due to ripening could be higher in year 1 than in year 2, i.e., most accessions grown in year 2 appear concentrated in the 1 to 2 conventional slope values, while accessions grown in year 1 appeared mostly around slope 2 or higher (Figure 5). On the contrary, the year did not appear to have an obvious effect on the increase in capsaicinoids during ripening, as most cases were found in the 1–100% increase (Figure 5).

## 3. Discussion

For decades, peppers have been identified as vegetables of high nutritional quality rich in biocompounds of claimed antioxidant properties, including high levels of minerals, vitamin C, carotenoids, and flavonoids [27,28,29], together with variable levels of capsaicinoids, low acids, and sugars [30,31,32].

This work focused on the levels of sugars and capsaicinoids as they influence the flavor of *Capsicum* peppers and the effect that the ripening and the environmental factors have on their accumulation. Although neglected, the important contribution of sugars in the organoleptic properties of peppers is well-known, as these primary metabolites are implicated the sweetness, aroma, and flavor attributes of fruits [33]. On the other hand, capsaicinoids are specific alkaloids related to the pungent attributes of pepper fruits and have been used in the instrumental Scoville Heat Test for the determination of pungency since the late 1970s [34]. There, the concentration of individual capsaicinoids allows us to determine the Scoville pungency, also considering the individual threshold pungencies according to the following order: capsaicin = dihydrocapsaicin > nordihydrocapsaicin > other minor capsaicinoids. Therefore, the added concentration of capsaicin and dihydrocapsaicin is the main determinator for pepper pungency [35].

The analyses separated data from the green-ripe and fully ripe stages. Ripening is a physiological factor that affects the composition of *Capsicum* peppers [23]. In general, sugars in fruits increase with ripening until they reach their physiological maturity [36]. In our study, levels of both fructose and glucose were around 1.8-fold higher than the green-ripe fruits. Similar or even higher ratios derived from the maturity of fruits have been determined by previous authors within the *Capsicum* complex [25,37,38]. On the other hand, sucrose was degraded during ripening which may be due to the action of the sucrose synthases, as firstly explained by Nielsen et al. [39], due to the carbohydrate metabolism during fruit development in sweet peppers. By contraposition, the effect of ripening was not as clear for capsaicinoids, though nearly all the accession showed higher capsaicinoid content at the fully ripe stage. Previous authors found a positive effect of ripening in the accumulation of capsaicinoids as well. During fruit development, capsaicinoids are increased until reaching maximum levels just before or at ripening [40,41,42]. After that, a period of stability or a decrease in capsaicinoids may be found in ripe fruits, the latter due to the degradation by the peroxidase activity [43], followed by a final over-maturation increase [42,44]. The accumulation of capsaicinoids in the placenta and seeds has been suggested as a kind of evolutionary defense of *Capsicum* plants in nature to protect their offspring against mammals at the most crucial stage (i.e., when achieving the physiological ripening, close to seed release) [32], and our findings confirm this hypothesis.

Within each ripening stage, the genotype was an important contributor to the accumulation of sugars and capsaicinoids in the studied *Capsicum* peppers, as found in the ANOVAs, especially the latter. Studies elucidating the biosynthesis of capsaicinoids have resulted in the identification of several genes, including Pun1, whose expression plays a key role in capsaicinoid production and the differential accumulation between non-pungent, pungent, and super-hot *Capsicum* peppers [45]. This genotypic control would explain the superiority of the *C. chinense* ECU994 in the present study over the *C. baccatum* and *C. annuum* accessions, despite the influence of the environmental conditions.

The levels found in the current study for both sugars and capsaicinoids are comparable to the literature [46], which found that the content of available carbohydrates (expressed as monosaccharides) in common fruits from the families *Solanaceae* and *Cucurbitaceae* ranged between 1.9 (cucumber) and 4.4 (eggplant) g 100 g^−1^, while other vegetables, such as beet or carrot, increased these levels to over 6.0 g 100 g^−1^. We found levels for the green-ripe fruits close to the levels determined by those authors for cucumber and squash, while fully ripe fruits in most accessions exceeded, for example, the average tomato contents or even reached the beet levels in some genotypes for year 1. Furthermore, the levels found in our work are in the range of previous studies for pepper considering either green-ripe or fully ripe fruits [17,20,47], although some authors found lower levels [33]. Contrarily, Rosa-Martínez et al. [48] found that fully ripe peppers under organic farming could reach total sugar levels over 90 g kg^−1^, although the average content was 67.2 g kg^−1^, thus indicating high variability within the *Capsicum* germplasm for the accumulation of sugars. We found overall lower levels, which may be affected by the environmental conditions and the agronomic practices, as well as the genotypes tested, thus reflecting the importance of comparing genotypes under similar conditions. Nevertheless, our values at the fully ripe stage were similar to those of the tomato (e.g., [49]) and, therefore, we cannot consider peppers as a low-in-sugar tasteless species, as reported historically [2]. On the contrary, these metabolites should be studied in breeding initiatives aimed at improving flavor, together with capsaicinoids (in hot peppers) and volatiles.

Additionally, most of the hot accessions evaluated in the present study showed total levels of capsaicinoids in the range of those determined by other authors for mild to intermediate pungent peppers. Bae et al. [8] found levels between 21 and 230 µg g^−1^ in several *C. annuum* accessions, which increased between 1.5-fold and 4.5-fold when peppers were fully ripe. Surprisingly, the same study by Giuffrida et al. [50] found higher levels in Jalapeño than in Serrano, contrary to our results. Such difference might correspond to the genotypes studied in each case, especially considering that there is pungent and non-pungent jalapeño germplasm (as it is the variety called the “TAM Mild Jalapeño”). Among capsaicinoids, capsaicin and dihydrocapsaicin were the main compounds found in all accessions. These two compounds are described as the most abundant capsaicinoids in pepper fruits, and represent around 75–95% abundance of this fraction in *Capsicum* spp. [44,51], which agrees with our results. Furthermore, the capsaicin:dihydrocapsaicin ratio determined here for Serrano was similar to the relative abundance found by Giuffrida et al. [50], thus suggesting great stability for this accession across environments. Additionally, our spiciest accession, ECU994, showed a capsaicin:dihydrocapsaicin ratio close to the levels for the also spicy Habanero Chocolate and Habanero Golden (~6), as discussed by Giuffrida et al. [50]. On the other hand, three out of the ten accessions lack capsaicinoids. During the domestication and breeding processes, sweet peppers were selected for the absence of capsaicinoids. Therefore, it is not surprising that the non-pungent accessions included in this work did not show quantifiable amounts of capsaicinoids, as described elsewhere [8,52,53,54].

Despite the importance of the genotype and ripening, the influence of the environmental conditions during cultivation is also to be considered in the accumulation of both sugars and capsaicinoids. Our study showed a high influence of the year of cultivation depending on the genotype, with an overall increase in year 1 that would be favored by the environmental conditions. In this sense, Kim et al. [55] also reported a significant effect of the year of cultivation for the accumulation of both sugars and capsaicinoids. Tripodi et al. [56] found a low but significant effect of the year of cultivation and the G × Y interaction in the accumulation of glucose and fructose in fully ripe sweet pepper landraces that explained around 3.2% of the variability each, which in our study reached 14.7% for the year factor and 29.7% for the G × Y interaction in fully ripe fruits. Surprisingly, BOL58 was poor in capsaicinoids in year 1 but it accumulated intermediate levels in year 2, thus showing relevant low stability. Overall, the differences in the chemical composition of *Capsicum* peppers due to the year and the G × Y interaction support the need of performing works of agronomic and nutritional characterization over several years as a tool to reduce the effect of this factor. This becomes especially important for crops developed in the open field, due to the lower control that can be exerted over the environmental conditions in this case.

Finally, the cultivation system did not influence the accumulation of the compounds studied, though significant interactions with the genotype were found, mainly for sugars, thus suggesting that most of these materials could be equally used for either organic or conventional cultivation. Overall, these low differences, with some exceptions due to the G × S interaction, suggest that crop management has a lower influence on the accumulation of primary metabolites and capsaicinoids compared to other factors, while it may influence the accumulation of antioxidant compounds [24] as a response to a differential abiotic stress. It should be noted that in our study, we selected two experimental fields close enough to minimize the environmental differences, thus they were mainly due to the cropping system. By contrast, other studies have reported that the location can exert greater effects, especially if they correspond to marked differential [56,57]. The results of these works suggest that the characterization and selection works, either for broad stability or specific for a considered cultivation system, should be tested across environments representing the conditions in which these materials will be used.

## 4. Materials and Methods

### 4.1. Plant Material and Growing Conditions

Eight *Capsicum annuum* accessions from different varietal types and origins, encompassing some Spanish traditional varieties and others from abroad, as well as one *C. baccatum* and one *C. chinense* accession, were evaluated in this work (Table 3 and Figure 6). Plants were cultivated in an open field in the area of Sagunto (Valencia, Spain), under two different growing conditions, organic and conventional cultivation, during the spring–summer cycle (from April to September) of the years 2016 and 2017, i.e., year 1 and year 2, respectively, whose temperature regime is displayed in Appendix A. The organic and conventional fields were close to each other (UTM coordinates X: 734,494.88 and Y: 4,390,434.86 in the organic plot and X: 732,900.40 and Y: 4,391,754.37 in the conventional plot) so that plants were grown under similar environmental conditions in terms of climate, main soil properties, and irrigation.

Ten plants per accession and growing system were transplanted at the four true leaves stage, distributed in five blocks of two plants each. The plot was divided into five rows and all the accessions were represented by one block within each row following a randomized design within the row, with a planting frame of 1.0 × 0.5 m between rows and within the row, respectively. Additionally, plants remaining after transplanting were used as borders to achieve two rows of plants surrounding each side of the experimental plot. Growing conditions were as described in Ribes-Moya et al. [23,24]. Briefly, surface irrigation was used every 8 to 10 days, depending on evapotranspiration. For the organic system, organic sheep manure (4 kg m^−2^) was spread as fertilizer at the beginning of the season. Pest control was not necessary because of the presence of natural predators, and weeds were monthly removed by hand. For conventional cultivation, one application of a mixture of inorganic nitrogen, phosphorus, and potassium (15-15-15) (50 g m^−2^) was conducted prior to the transplant. Additional applications of calcium nitrate and iron chelate were conducted during the growing cycle. Pests were controlled by applying chlorpyrifos (48%, EC) and abamectin (1.8%, EC), and copper oxychloride (58.8% WP) was used as a fungicide. Weeds were controlled by hand as in the organic system.

### 4.2. Sampling

Fruits for each accession were harvested at the two main commercial stages of peppers: (a) green-ripe (final size and firm fruit, green) and (b) fully ripe (final size and firm fruit, completely colored with carotenoids). Each sample was prepared with fruits of the two plants from each block (i.e., 1 block = 1 sample) and, therefore, five biological replicates (*n* = 5) per accession × growing condition × ripening stage combination were prepared and analyzed each year. Thus, a total of 400 samples were prepared and analyzed in the present work. Samples were prepared by cutting the fruits into small pieces and then 30 g (fresh weight) of each sample was lyophilized. The weight before and after lyophilization was recorded to estimate the dry matter/fresh matter ratio of each sample. The lyophilized samples were ground to a fine particle size using a coffee grinder, and preserved in darkness and dry conditions until the analyses were made.

### 4.3. Chemical Determinations

For the sugar determinations, 0.100 g of lyophilized material was diluted in 1.5 mL of ultrapure water. The samples were extracted with a vortex agitator set at maximum rpm rotational speed for 1 min, and centrifugated for 5 min at 12,000 rpm. The supernatant was filtered through 0.22 µm PVDF syringe filters and analyzed in an Agilent 1220-Infinity HPLC System (Agilent Technologies, Santa Clara, CA, USA) equipped with an RI detector (Varian Pro Star, model 350, Palo Alto, CA, USA). The compound separation was performed with a Luna^®^ Omega SUGAR LC column (150 mm × 4.6 mm i.d., 3 µm particle size) (Phenomenex, Torrance, CA, USA) and a guard column (SUGAR, 4 mm × 3.0 mm i.d.). The mobile phase consisted of 25:75 *v/v* water and acetonitrile set at a flow rate of 1 mL min^−1^ for 15 min. The injection volume was 10 µL, and the column temperature was set to 35 °C during analysis. Quantifications were based on calibration curves prepared for glucose, fructose, and sucrose by the external standard method from 1 g L^−1^ to 10 g L^−1^ [48].

Capsaicinoids were determined as described by Sganzerla et al. [58] and Barberó et al. [59] with slight modifications. Thus, 0.100 g of the lyophilized samples were homogenized with 1.5 mL of methanol and capsaicinoids were extracted by ultrasound-assisted extraction at 50 °C for 10 min at 40 kHz of ultrasonic frequency [59]. The samples were centrifugated at 5000 rpm for 2.5 min and filtered through 0.22 µm PTFE syringe filters and analyzed by HPLC-UV in an Agilent 1220-Infinity HPLC System (Agilent Technologies) at 280 nm. Individual compounds were separated using a Luna^®^ C18 POLAR LC column (150 mm × 4.6 mm i.d., 3 µm particle size) (Phenomenex). The mobile phase consisted of 60:40 *v/v* acetonitrile and water containing 0.1% acetic acid in the isocratic mode set at a flow rate of 0.8 mL min^−1^ for 20 min [58]. The injection volume was 10 µL and the column temperature was set to 40 °C during analysis. Quantifications were based on external calibration obtained by serial dilutions of commercial standards of capsaicin and dihydrocapsaicin covering the concentration ranges from 0.1 to 500 ppm.

### 4.4. Statistical Analysis

Data were analyzed using Statgraphics Centurion XVIII (Statgraphics Technologies, The Plains, VA, USA). The significance of the effects genotype (G), growing system (E, environment), ripening stage (R), and year (Y) and their interactions were tested with Analysis of Variance/ANOVA at different levels (two- and three-way ANOVAs). Post-hoc comparisons were performed using the Student–Newman–Keuls (SNK) test at *p* < 0.05.

## 5. Conclusions

In the present work, we have assessed comprehensively, based on a range of varietal types, how the genotype, the ripening stage, the growing system, the year, as well as their interactions contribute to the variation in capsaicinoids and sugars in *Capsicum* peppers. Our study found a remarkable genotypic variation in both traits, with a particular interest in sugars, with several varieties reaching values similar to those of tomatoes or carrots at the fully ripe stage (30–50 g 100 g^−1^ of fw) and, therefore, we confirmed this trait as relevant for quality breeding programs of peppers. The ripening stage had also a high impact on the levels of capsaicinoids and sugars, being higher at the fully ripe stage in most accessions (on average 96 vs. 143 mg kg^−1^ in total capsaicinoids and 20 vs. 37 g kg^−1^ in total sugars). The ripening stage considerably changed the profile of sugars (considerable levels of sucrose, fructose, and sucrose in green-ripe fruits vs. a lack of sucrose and balanced percentages of sucrose and fructose in fully ripe peppers). The year also had a significant effect, while the growing system had only minor effects. Finally, G × E and G × Y interactions also contributed significantly to the variation of both traits.

## Figures and Tables

**Figure 1 plants-12-00231-f001:**
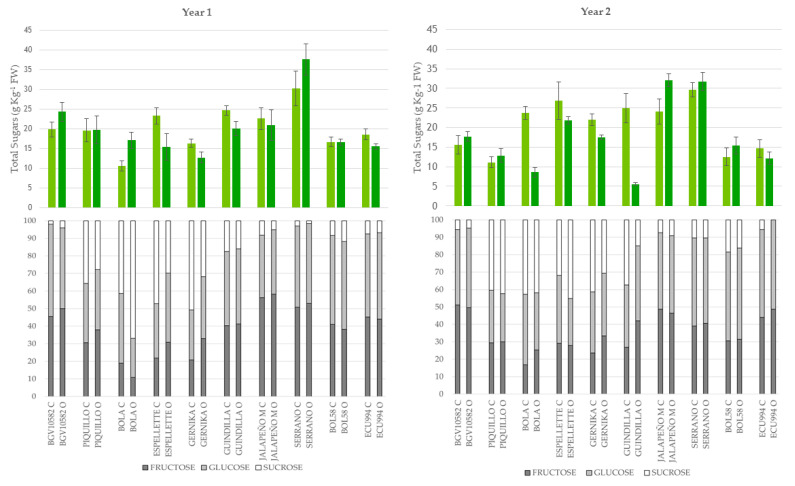
Content of total sugars (upper, g kg^−1^ of fresh weight) and sugar profile (lower, % of total sugars, grey = fructose, pale grey = glucose, and white = sucrose) at the green-ripe stage for the accessions evaluated under conventional (pale green or C) and organic (green or O) growing conditions in years 1 and 2. Vertical bars in the upper figures indicate SE intervals for each mean value.

**Figure 2 plants-12-00231-f002:**
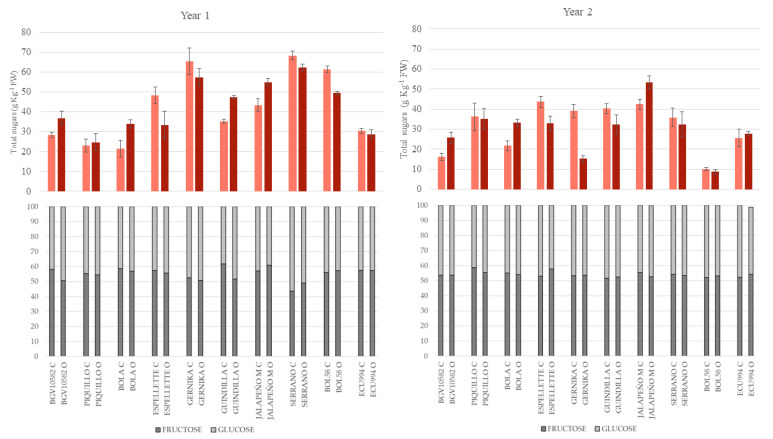
Content of total sugars (upper, g kg^−1^ fresh weight) and sugar profile (lower, % of total sugars, grey = fructose, and pale grey = glucose) at the fully ripe stage for the accessions evaluated under conventional (pale red or C) and organic (red or O) growing conditions in years 1 and 2. Vertical bars in the upper figures indicate SE intervals for each mean value.

**Figure 3 plants-12-00231-f003:**
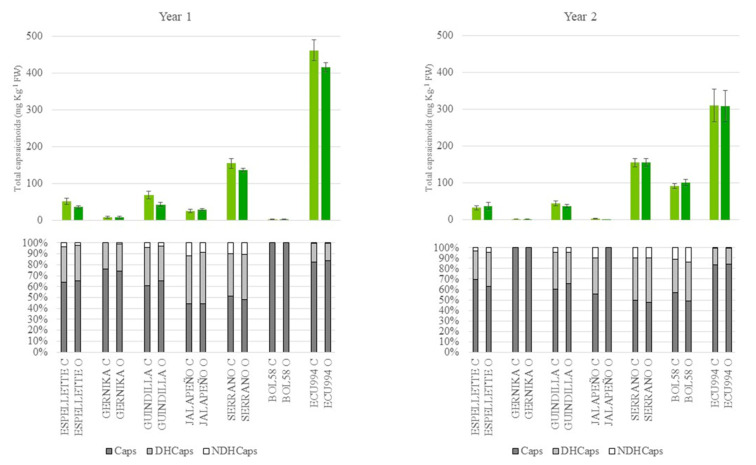
Content of total capsaicinoids (upper, mg kg^−1^ of fresh weight) and capsaicinoid profile (lower, % of total capsaicinoids, grey = capsaicin, pale grey = dihydrocapsaicin, and white = nordihydrocapsaicin) at the green-ripe stage for the accessions evaluated under conventional (pale green or C) and organic (green or O) growing conditions in years 1 and 2. Vertical bars in the upper figures indicate SE intervals for each mean value.

**Figure 4 plants-12-00231-f004:**
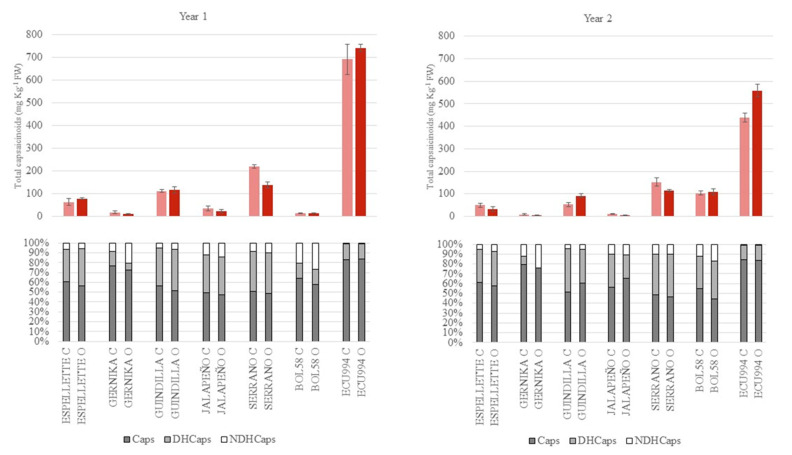
Content of total capsaicinoids (upper, mg kg^−1^ of fresh weight) and capsaicinoid profile (lower, % of total capsaicinoids, grey = capsaicin, pale grey = dihydrocapsaicin, and white = nordihydrocapsaicin) at the fully ripe stage for the accessions evaluated under conventional (pale red or C) and organic (red or O) growing conditions in years 1 and 2. Vertical bars in the upper figures indicate SE intervals for each mean value.

**Figure 5 plants-12-00231-f005:**
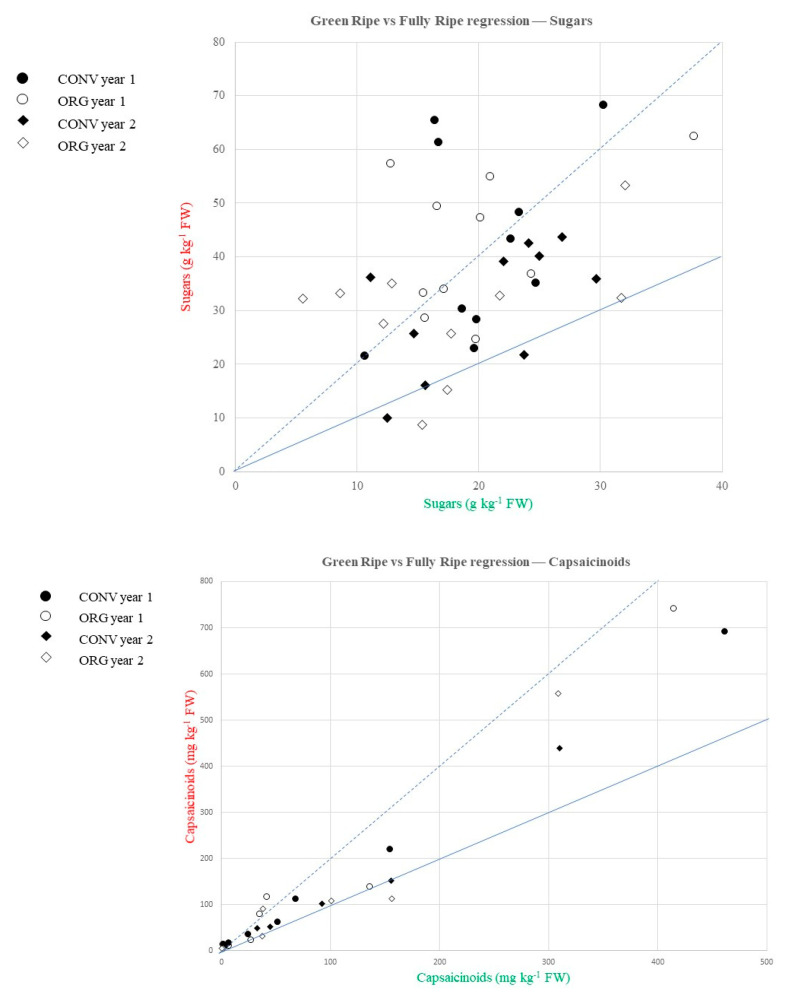
Green-ripe (green) vs. fully ripe (red) means of the studied *Capsicum* collection for total sugars (upper, g kg^−1^ of fw) and total capsaicinoids (bottom, mg kg^−1^ of fw) in both years and growing systems. The continuous blue line indicates slope = 1 (i.e., the same value at the green-ripe than the fully ripe stage). The dotted blue line indicates slope = 2 (i.e., the value at the fully ripe stage is 2-fold greater than at the green-ripe stage).

**Figure 6 plants-12-00231-f006:**
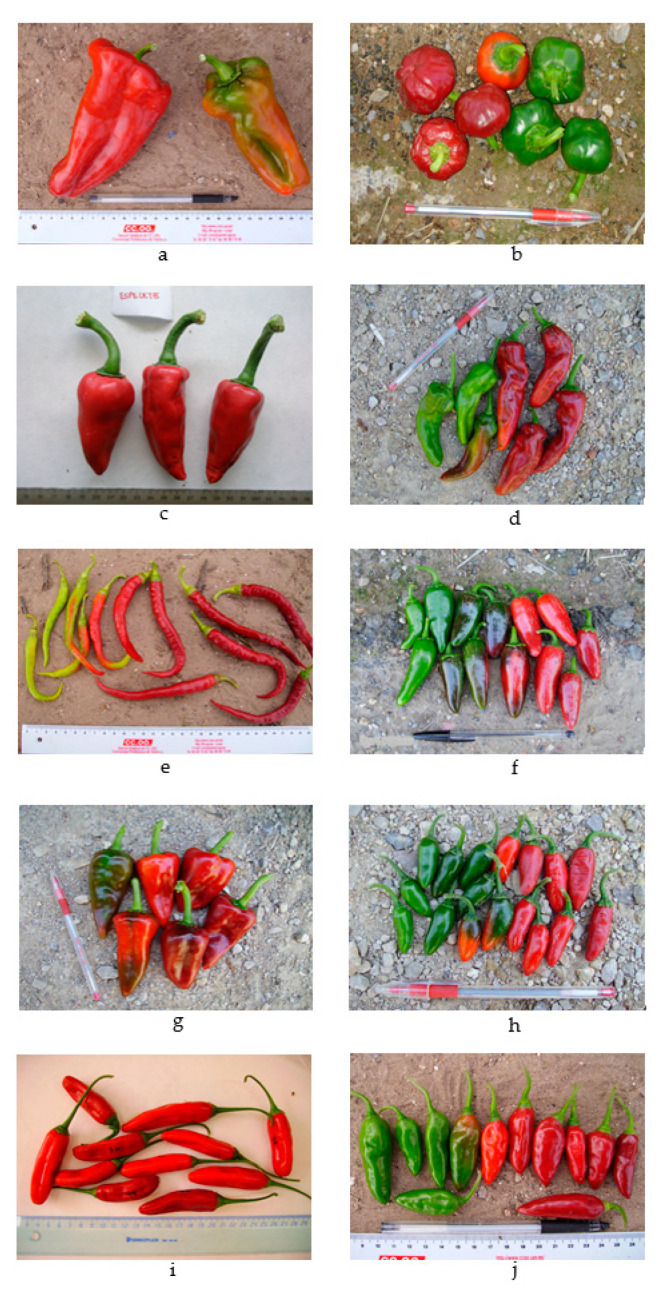
Fully ripe fruits of the accessions used. Rulers at the bottom are in cm (marks between two numbers indicate 1 cm). The pen has a 150 mm length. In pairs, from left to right, and in descending direction: BGV10582 (**a**), Bola (**b**), Espelette (**c**), Gernika (**d**), Guindilla de Ibarra (**e**), Jalapeño (**f**), Piquillo (**g**), Serrano (**h**), *C. baccatum* BOL58 (**i**), and *C. chinense* ECU994 (**j**).

**Table 1 plants-12-00231-t001:** ANOVA’s total sum of squares (expressed in %) for the effects the genotype (G), growing system (E), ripening stage (R), and year (Y), and their interactions for the content in sugars: fructose, glucose, sucrose and total sugars, and capsaicinoids: capsaicin (Caps), dihydrocapsaicin (DHCaps), nordihydrocapsaicin (NDHCaps), and total capsaicinoids.

	Sugars									Capsaicinoids								
EFFECT	df	Fructose		Glucose		Sucrose		Total		df	Caps		DHCaps		NDHCaps		Total	
Genotype (G)	9	14.65	***	19.24	***	15.07	***	18.18	***	6	83.01	***	74.19	***	65.65	***	81.20	***
Growing system (E)	1	0.00	ns	0.10	ns	0.92	***	0.17	ns	1	0.01	ns	0.01	ns	0.00	ns	0.00	ns
Ripening stage (R)	1	45.81	***	30.70	***	31.74	***	30.12	***	1	1.17	***	2.36	***	3.16	***	1.55	***
Year (Y)	1	4.84	***	3.40	***	0.02	ns	4.83	***	1	0.65	***	0.61	***	0.29	**	0.65	***
Interactions																		
G × E	9	2.34	***	3.09	***	2.56	***	4.19	***	6	0.44	**	0.64	**	1.74	***	0.47	**
G × R	9	4.11	***	4.27	***	15.10	***	2.80	***	6	4.42	***	3.28	***	1.43	***	4.33	***
G × Y	9	7.91	***	10.87	***	1.34	ns	10.83	***	6	4.17	***	9.29	***	17.97	***	5.20	***
E × R	1	0.02	ns	0.03	ns	0.93	***	0.06	ns	1	0.07	ns	0.03	ns	0.01	ns	0.07	ns
E × Y	1	0.03	ns	0.10	ns	0.09	ns	0.11	ns	1	0.07	ns	0.20	*	0.18	*	0.11	*
R × Y	1	2.30	***	2.96	***	0.02	ns	3.22	***	1	0.18	**	0.42	***	0.39	**	0.24	**
Residual		18.00		25.23		33.49		25.50			5.79		8.98		9.18		6.18	

df: degree freedom. ns, *, **, and *** indicate non-significant or significant at *p* < 0.05, 0.01, and 0.001 in the Student–Newman–Keuls test, respectively.

**Table 2 plants-12-00231-t002:** ANOVA’s total sum of squares (expressed in %) for the effects of the genotype (G), growing system (E), and year (Y), and their interactions for the content in sugars: fructose, glucose, sucrose, and total, and capsaicinoids: capsaicin (Caps), dihydrocapsaicin (DHCaps), nordihydrocapsaicin (NDHCaps), and total, considering separately green-ripe and fully ripe stages.

	Sugars									Capsaicinoids								
EFFECT	df	Fructose		Glucose		Sucrose		Total		df	Caps		DHCaps		NDHCaps		Total	
	Green-ripe stage																	
Genotype (G)	9	69.92	***	51.01	***	43.52	***	41.07	***	6	50.18	***	82.03	***	77.38	***	88.97	***
Growing system (E)	1	0.13	ns	0.64	ns	2.49	**	1.04	*	1	0.37	ns	0.11	ns	0.01	ns	0.08	ns
Year (Y)	1	1.70	***	0.02	ns	0.04	ns	0.44	ns	1	1.57	**	0.03	ns	1.38	***	0.18	*
Interactions																		
G × E	9	2.55	*	6.26	***	7.66	**	9.81	***	6	0.30	ns	0.64	ns	0.60	*	0.13	ns
G × Y	9	6.37	***	9.34	***	4.06	ns	9.53	*	6	21.71	***	10.81	***	16.00	***	5.61	***
E × Y	1	0.11	ns	0.49	ns	0.30	ns	0.65	ns	1	0.26	ns	0.24	*	0.15	ns	0.09	ns
Residual		19.20		32.24		41.92		37.46			25.61		6.13		4.48		4.94	
	Fully ripe stage																	
Genotype (G)	9	25.07	***	29.55	***	ND		27.10	***	6	88.03	***	78.34	***	61.04	***	86.57	***
Growing system (E)	1	0.04	ns	0.02	ns	ND		0.03	ns	1	0.07	ns	0.00	ns	0.01	ns	0.06	ns
Year (Y)	1	16.60	***	11.71	***	ND		14.67	***	1	0.99	***	1.56	***	0.01	ns	1.12	***
Interactions																		
G × E	9	8.18	***	7.73	***	ND		7.97	***	6	1.34	***	2.56	***	5.01	***	1.54	***
G × Y	9	29.16	***	30.40	***	ND		29.66	***	6	4.71	***	9.57	***	21.78	***	5.53	***
E × Y	1	0.07	ns	0.11	ns	ND		0.09	ns	1	0.09	ns	0.19	ns	0.23	ns	0.12	ns
Residual		20.88		20.49				20.48			4.76		7.78		11.92		5.07	

df: degrees of freedom. ns, *, **, and *** indicate non-significant or significant at *p* < 0.05, 0.01, and 0.001 in the Student–Newman–Keuls test, respectively. ND: not detected in fully ripe fruits.

**Table 3 plants-12-00231-t003:** List of accessions used, including their local name, origin, and some relevant fruit traits.

Accession	Origin	Color	Mesocarp	Shape (Pochard’s)	Length/Width (mm)	Weight (g)
*C. annuum*						
BGV10582	COMAV, Valencia, Spain	Red	Thick, fleshy	Bell elongated (B1)	162/73	>250
Bola	PDO Pimenton de Murcia, Totana, Spain	Deep red	Thin, high, and dry matter	Round (N)	35/41	10–25
Espelette	F. Jourdan. INRA Geves, France	Deep red	Thin, high, and dry matter	Elongated (C3)	138/29	25–50
Gernika	Neiker, Euskadi, Spain	Deep red	Thin, high, and dry matter	Elongated (C2)	84/32	25–50
Guindilla	Neiker, Euskadi, Spain	Red	Thin, high, and dry matter	Very elongated (C1)	139/11	<10
Jalapeño	USA. Reimer Seeds	Deep red	Thick, fleshy	Elongated (B4)	65/31	10–25
Piquillo	PDO Piquillo, Navarra, Spain	Deep red	Medium	Triangular (C4)	94/48	50–100
Serrano	Mexico. Reimer’s Seeds Co.	Red	Medium	Elongated (B4)	35/16	<10
*C. baccatum*						
BOL58	Cochabamba (Bolivia)	Deep red	Thin, high, and dry matter	Elongated	70/15	5–10
*C. chinense*						
ECU994	Archidona, Napo (Ecuador)	Red	Thin, high, and dry matter	Triangular	42/15	5–10

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
