# Peer review of "The Effect of the Varietal Type, Ripening Stage, and Growing Conditions on the Content and Profile of Sugars and Capsaicinoids in Capsicum Peppers"

_plants, 2023, doi:10.3390/plants12020231_

Round 1
Reviewer 1 Report
Review of MS 'Effect of the varietal type, ripening stage and growing conditions on flavor traits of Capsicum peppers: sugars and capsaicinoids profiles.
Title: does not correspond to the topic of the manuscript! It should be rewritten. Flavour traits are determined by several quality parameters, including organic acids, which were not discussed in this study; growing conditions are very poorly described in this MS;
Introduction: Authors reported that developmental stage of the fruits is known to affect the accumulation of metabolites, which increase to an optimal stage at the ...... (Line 81). What is the optimal stage of the metabolites? The description of the growing conditions regarding the climate conditions is missing if authors discuse the influence of growing conditions to chemical profile of Capsicum pepper fruit.
Material and Methods:
Important drawback: the experiment consisted of 10 plants per accession and cropping system. There were no field replicates in this study, which is inadmissible for studying the effects of environmental conditions and genotype on fruit quality parameters in a field experiment. Ten plants are far too few for this kind of experiment, in which the influence of so many factors is to be evaluated!!
The experiment is inadequately described. The growing conditions are inadequately described. If the authors compared two cropping systems, then the soil properties should be described and the agrotechnical measures should be discussed in more detail.
Statistical analysis: For the analysis of the chemical composition of sugars as well as the chemical compostions of capsaicinoides, the data should be analysed using the compositionsal data analysis.
(See: Aitchinson, J. The statistical analysis of compositional data; The Blackburn press: Caldwell, NJ, 2003; 408 pp.; Pawlowsky-Glahn, V.; Egozcue, J. J. Geometric approach to statistical analysis on the simplex. Stochastic Environ. Res. Risk Asses. (SERRA) 2001, 15 (5), 384–398).
Results:The author compared the fruit quality in two ripeness stages: green-ripe and fully ripe; the results of the colour measurements are missing. Please add them.
Author Response
Dear Reviewer 1,
attached please find our changes and answers to your suggestions and concerns. We are grateful for your time and suggestions, which have contributed significantly to improve the MS.
With sincere regards
The authors

Reviewer 2 Report
We suggest to the authors:
On Figure 6 - for better and faster identification of the figures, please use letters to identify each photo example "a)", "b)"...
On the values described on the text "19.55 g kg-1 fresh 148 weight (fw)", please use for the rest of the text also "fw" when needed.
Please clarify in ln 507 "Ten plants per accession and growing system were transplanted at the 4 true leaves stage following a randomized design with planting frame of 1.0 × 0.5 m between rows and within row, respectively. Growing conditions were as described in Ribes-Moya et al. [24]" Your text should contain what randomized block design you have used, number of replications, size of the plot and borders existence or not. Even that some information could be in [24] this is very important to be missed here.
Author Response
Dear Reviewer 2,
attached please find our changes and answers to your suggestions and concerns. We are grateful for your time and suggestions, which have contributed significantly to improve the MS.
With sincere regards,
The authors

Reviewer 3 Report
In the article you talk about: varietal type, varietal diversity, accession
I observed you have worked with 3 species. Within the species there are accessions codified according to the international nomenclature (ex ECU994) but for C. Annum they are local varieties?
I suggest you review all the article and insert the correct nomenclature
Considering that you have worked on 2 years and that there are seasonal effects, it would be useful to at least present the temperature data during the cultivation cycle in the 2 years.
Line 19: farming system substitute with growing system and insert (E) because al line 22 you indicate GxE
Line 24: insert from before 97 and on average after Kg-1 like line 23
Line 25 replace – with and
Line 29: delete breeding
Line 53: delete t
Line 66: Dw insert (dry weight)
Line 88: replace growing condition with growing system
Line 88-91: rewrite in a simpler way
Line 118: indicate what is TSS
Line 116-124 is not indicated if you are referring to green ripe stage or fully ripe stage?
Line 179: insert fig or tab references
Line 510: have you measured evapotraspiration?
2.4 rewriting: did not assess the maturation process but only two steps
Unripe probably is green ripe.
Fig. 5 in the figure and in the caption substitute unripe with green ripe.
it is not very clear to me how you did the regression. you plotted pairs of values on samples at different stages of ripening of different fruits. can you give me a clarification?
Fig. 6 . In the caption you indicate only full ripe. But there are also green ripe. The picture are not good quality. You cloud try to insert letters for single pictures.
Author Response
Dear Reviewer 3,
attached please find our changes and answers to your suggestions and concerns. We are grateful for your time and suggestions, which have contributed significantly to improve the MS.
With sincere regards
The authors

Reviewer 4 Report
In this manuscript by Guijarro-Real et al., the authors studied the content and profile of major sugars and capsaicinoids in ten different genotype peppers, other factors including: different farming systems, different years and the two main ripening stages. The results showed that ripening stage and the genotype are the major contribution to the content of sugars and capsaicinoid. And they found that in fully ripe fruits in several accessions reaching total sugars similar to tomatoes. The data presented could be potentially valuable for pepper breeding. There are some issues which need to be clarified:
1. Too many keywords in this manuscript, 4-6 are suitable.
2. There are many formatting errors in this manuscript, e.g. line 53: “t” is redundant; line 149: missing a period? Besides that, the writing of the units in this manuscript is not standardized, dot is missing in the middle, please check it.
3. The format of references should be uniform, for example, the first letter of the quoted title should accordance, most of them are lower case, while reference 21 is capital. The author's name of the reference needs to be unified, e.g. reference 3, 18, 27, 28, 30.
Author Response
Dear Reviewer 4,
attached please find our changes and answers to your suggestions and concerns. We are grateful for your time and suggestions, which have contributed significantly to improve the MS.
With sincere regards
The authors

Round 2
Reviewer 1 Report
Dear Editor,
I have reviewed the revised version of the manuscript plants2084534 entitled Effect of the varietal type, ripening stage ad growing conditions on flavour traits of Capsicum peppers: sugars and capsaicionids profiles. I thank the authors who corrected the MS according to my suggestions and added appropriate comments to all suggestions not considered.